# Circulating MicroRNAs Highly Correlate to Expression of Cartilage Genes Potentially Reflecting OA Susceptibility—Towards Identification of Applicable Early OA Biomarkers

**DOI:** 10.3390/biom11091356

**Published:** 2021-09-13

**Authors:** Yolande F. M. Ramos, Rodrigo Coutinho de Almeida, Nico Lakenberg, Eka Suchiman, Hailiang Mei, Margreet Kloppenburg, Rob G. H. H. Nelissen, Ingrid Meulenbelt

**Affiliations:** 1Department of Biomedical Data Sciences, Section Molecular Epidemiology, Leiden University Medical Center, 2333 ZC Leiden, The Netherlands; R.Coutinho_de_Almeida@lumc.nl (R.C.d.A.); N.Lakenberg@lumc.nl (N.L.); H.E.D.Suchiman@lumc.nl (E.S.); i.meulenbelt@lumc.nl (I.M.); 2Sequence Analysis Support Core, Leiden University Medical Center, 2333 ZC Leiden, The Netherlands; H.Mei@lumc.nl; 3Department of Rheumatology, Leiden University Medical Center, 2333 ZA Leiden, The Netherlands; G.Kloppenburg@lumc.nl; 4Department of Orthopaedics, Leiden University Medical Center, 2333 ZA Leiden, The Netherlands; R.G.H.H.Nelissen@lumc.nl

**Keywords:** circulating microRNAs, osteoarthritis, biomarker

## Abstract

Objective: To identify and validate circulating micro RNAs (miRNAs) that mark gene expression changes in articular cartilage early in osteoarthritis (OA) pathophysiology process. Methods: Within the ongoing RAAK study, human preserved OA cartilage and plasma (*N* = 22 paired samples) was collected for RNA sequencing (respectively mRNA and miRNA). Spearman correlation was determined for 114 cartilage genes consistently and significantly differentially expressed early in osteoarthritis and 384 plasma miRNAs. Subsequently, the minimal number of circulating miRNAs serving to discriminate between progressors and non-progressors was assessed by regression analysis and area under receiver operating curves (AUC) was calculated with progression data and plasma miRNA sequencing from the GARP study (*N* = 71). Results: We identified strong correlations (ρ ≥ |0.7|) among expression levels of 34 unique plasma miRNAs and 21 genes, including 4 genes that correlated with multiple miRNAs. The strongest correlation was between let-7d-5p and *EGFLAM* (ρ = −0.75, *P* = 6.9 × 10^−5^). Regression analysis of the 34 miRNAs resulted in a set of 7 miRNAs that, when applied to the GARP study, demonstrated clinically relevant predictive value with AUC > 0.8 for OA progression over 2 years and near-clinical value for progression over 5 years- (AUC = 0.8). Conclusions: We show that plasma miRNAs levels reflect gene expression levels in cartilage and can be exploited to represent ongoing pathophysiological processes in articular cartilage. We advocate that identified signature of 7 plasma miRNAs can contribute to direct further studies toward early biomarkers predictive for progression of osteoarthritis over 2 and 5 years.

## 1. Introduction

As stated by Peat and Thomas, the importance of osteoarthritis (OA) to population health and health systems is more and more recognized; however, the position of OA as a leading cause of disability worldwide is still undervalued [1]. The relevance of this recognition is emphasized by the fact that, globally, both prevalence and years lived with disability (YLD) for OA have increased by almost 10% in the past 20 years [2]. Guidelines and recommendations are provided and regularly updated in view of newly gathered knowledge on OA aetiology to direct decision-making for disease management of clinicians and patients [3]. However, despite these records, as of yet, no treatment to cure and/or slow down OA is available. For that matter, the lack of sensitive and objective clinical markers with potential to serve in OA prediction, diagnosis, and prognosis as well as to monitor disease over time in drug development and clinical trials has been a major drawback.

Most studied biomarkers are biochemical degradation products of cartilage or bone, such as serum cartilage oligomeric matrix protein (sCOMP) and urinary *C*-telopeptide of type I collagen (uCTXI) [4,5,6]. Nonetheless, these markers are a-specific and do not mark early OA. Moreover, their predictive value with area under the curve (AUC) of below 0.7 value, and only slightly different from covariates such as sex and body mass index (BMI) alone [7], has not reached clinical relevant levels, widely used as a threshold to indicate adequate discrimination performance (AUC ≥ 0.8) [8,9]. In this respect, advancing data of circulating non-coding RNAs (ncRNAs) such as micro RNAs (miRNAs), hold great promise as an effective tool to mark underlying disease processes and could sensitively monitor the effect of treatment options [10,11].

MiRNAs are secreted in the circulation where they directly reflect ongoing cellular and/or tissue processes and may signal to distant tissues. They are protected from RNase activity by virtue of their association with secreted membrane vesicles or RNA-binding proteins. Both aspects (signalling and stability) make miRNAs attractive targets as molecular biomarkers [12,13]. The first ncRNA with potential predictive value for severe knee or hip OA was let-7e, identified in 2014 by microarray screening [14]. In the years following identification of the first predictive OA miRNA, more analyses have been performed that added circulating miRNAs with potential value as biomarkers, such as miR-140-3p, miR-33b-3p, and miR-671-3p, marking the radiographic severity of OA [15,16]. While clinical and radiological examination is commonly used for diagnosis of OA, early OA pathophysiology is reflected by changes in gene expression in human articular cartilage even before it becomes apparent on radiographs. Nevertheless, to our knowledge, no previous studies have been reported on the relation of circulating miRNA with transcriptomic changes in OA cartilage. Knowing the urgency for biomarkers reflecting early changes in OA pathophysiology, we set out to identify miRNAs specifically marking such early gene expression changes in human articular cartilage. To this end, previously published datasets of genes differentially expressed with OA were exploited to select for those genes marking early OA independent of joint side [17,18,19]. Following miRNA sequencing of plasma collected within the RAAK study [20], miRNA expression levels were integrated with levels of selected early genes in articular cartilage from the same patients to identify circulating microRNAs with clinical predictive value for progression of OA over 2 and 5 years.

## 2. Materials and Methods

### 2.1. Sample Description

In the current study, plasma samples of 22 OA patients from the RAAK study (Research Arthritis and Articular Cartilage [20]) and 71 plasma samples of the GARP study (Genetics osteoARthritis and Progression [21]) were included. Ethical approval for both studies was obtained from the medical ethics committee of the Leiden University medical Center (RAAK: P08.239 and P19.013; GARP: P76.98), and informed consent was obtained from all participants.

Within the GARP study, radiographs of the hips and knees were obtained from participants at baseline and after both 2 and 5 years follow-up, while employing a standard protocol with a fixed film focus distance (1.15 m). Radiographs were scored blinded in known-time order by trained clinicians (radiologist, rheumatologist) according to the OARSI Atlas [22] for joint space narrowing (JSN, 0–3) and osteophytosis (OP, 0–3), as described before [23]. Increase in total OARSI score (sum of hips and knees) of more than 2 for osteophytosis (OP) and/or joint space narrowing (JSN) within 2 years (33 progressors) or 5 years (31 progressors) was defined as progression in our analyses.

### 2.2. Small RNA-Sequencing

Isolation of small RNAs was performed as described before [19], using 200 µL plasma. In short, Qiagen miRNeasy Serum/Plasma Kit (Qiagen, GmbH, Hilden, Germany) was used following the manufacturer’s protocol. Samples were sequenced in two batches. For the first batch including all GARP samples and 9 of the RAAK samples, small RNA sequencing libraries were constructed using the TruSeq rapid SBS kit (Illumina, San Diego, CA, USA). RNAs were separated on 4–20% SDS-PAGE and eluted from the gel to enrich for the pool of miRNAs. Small RNA-sequencing libraries for the second batch (13 RAAK samples) were constructed using NEBNext Small RNA Library Prep Set for Illumina (New England Biolabs GmbH, Leiden, The Netherlands) followed by BluePippin purification of the smallRNA fraction (Sage Science, Ochten, The Netherlands). After standard quality control (2100 Bioanalyzer RNA integrity Number (RIN) > 7), sequencing was performed, respectively, on the Illumina HiSeq 4000 and NovaSeq 6000 PE150 (Illumina, San Diego, CA, USA), yielding a mean of 11 million reads per sample. Adapters were removed using Cutadapt v1.1 [24] with 15 bp as a minimum length to keep after clipping. Small RNA-Seq data were aligned to the GRCh38 human reference genome with the software Bowtie version 1 [25] using best strata option. Read abundances were done with HTseq [26] and were further assigned with miRBase v22 [27]. In total, 2652 mature sequences annotated miRBase miRNAs could be mapped to the genome.

To generate a robust dataset for downstream analyses, a threshold of 8 reads was taken prior to normalization and transformation for generated miRNAseq datasets simultaneously using Variance Stabilizing Transform (VST) method from DESeq2 R package [28], and potential batch effects were removed using the removeBatchEffect function from the limma R package v 3.36.1 [29]. Subsequently, the upper expression quartile with VST-values of ≥2.8 in at least 50% of all samples was selected for downstream analyses (399 miRNAs).

### 2.3. mRNA-Sequencing Dataset

In this study, a subset of previously generated mRNA-sequencing data was included of macroscopically preserved OA cartilage from OA patients that had undergone total joint replacement surgery (*N* = 22, overlapping with individuals for which miRNA sequencing of plasma was performed) and for which sample characteristics have been described [19]. In short, strand-specific RNA-Seq libraries were generated prior to sequencing on Illumina HiSeq 2000 and Illumina HiSeq 4000 platforms, yielding a mean of 20 million reads per sample. Subsequently, RNA-sequencing reads were aligned using GSNAP [30] against GRCh38 using default parameters. Read abundances per sample was estimated using HTSeq count [26] while correcting for batch effects using the removeBatchEffect function from the limma R package v 3.36.1. Only uniquely mapping reads were used to estimate expression.

### 2.4. Analysis of Protein Interaction Networks

To explore for protein–protein interactions among cartilage genes correlating to plasma miRNAs, we used the Search Tool for the Retrieval of Interacting Genes/Proteins (STRING) 9.0 [31] available online (note: data are retrieved from source “previous knowledge” based on text-mining). STRING also allows for analyses of enrichment in gene functions, performed here for “GO Biological Processes”.

### 2.5. MicroRNA-mRNA Target Identification

To check whether any of the cartilage genes with strongest correlation to plasma miRNAs (21 genes and 34 miRNAs) were previously predicted and/or validated miRNA target genes, three prediction tools using the default parameters (DIANA-microT.v5 [32], miRDB.v6 [33], and TargetScan.v7.2 [34]), and two experimentally validated databases (miRTarBase.v7 [35] and TarBase.v8 [36]) were integrated using the multiMiR.v1.12.0 R package.

### 2.6. Calculation Area under Receiver Operating Curves (AUC)

Using Z scores generated for expression levels of 34 plasma miRNAs, classification models were constructed using multiple, penalized logistic regression for progression scores over 2 years, as described before [37]. Calculated coefficients were used to identify miRNAs characterizing progression over 2 or over 5 years (Appendix A). Subsequently, regression coefficients were calculated for sex, age, and BMI alone or while including expression levels of the panel of identified miRNAs by performing generalized estimation equations. These regression coefficients were used to generate receiver operator curves (ROC) for non-progressors versus progressors in parallel with regression coefficients while only including sex, age, and BMI.

Interpretation of AUC values: 0.50–0.59: no discrimination; 0.60–0.69: poor discrimination; 0.70–0.79: acceptable discrimination; and ≥0.80: excellent discrimination.

### 2.7. Statistical Methods and Analyses

Spearman correlations between plasma miRNAs (*N* = 399) and cartilage mRNAs (*N* = 114) were calculated in R statistical language while including 22 OA patients. Correlations were considered weak-to-moderate for |0.5| < ρ < |0.7| and strong for ρ ≥ |0.7| [38].

## 3. Results

### 3.1. Study Characteristics

To identify circulating miRNAs reflecting early changes in the articular cartilage transcriptome, miRNA sequencing was performed for plasmas of OA patients (N = 22). Table 1A shows characteristics of study participants. The majority were female (18 out of 22). Average age and BMI of the participants were, respectively, 71 and 28, with 16 undergoing knee and 6 undergoing hip replacement surgery. To identify readily detectable OA biomarkers, N = 399 plasma miRNAs of the highest expression quartile were selected for analyses. First, we explored whether any of the miRNAs correlated with potential covariates age and body mass index (BMI; Appendix A). In total, 8 miRNAs were found to moderately correlate (|0.5| < ρ < |0.6|) with age and 7 miRNAs with BMI. The most significant correlation with age and BMI was found for miR-3173-5p (ρ = 0.65, *P* = 1.2 × 10^−3^) and miR-369-3p (ρ = −0.56, *P* = 1.3 × 10^−2^), respectively. Identified miRNAs (15 in total) were excluded from further analyses to avoid interference of age and BMI with the outcome being OA, resulting in a total of 384 miRNAs that were included in correlation analysis with cartilage genes.

### 3.2. Selection of Genes Marking Early OA in Articular Cartilage

To identify genes marking early changes of OA pathophysiology in relevant joint tissues (Figure 1), we first prioritized on *N* = 158 overlapping, previously reported, differentially expressed genes (DEGs) when comparing non-OA and preserved OA cartilage of knee (*N* = 1418 DEGs) [17] and hip (*N* = 998 DEGs) [18] joints. To further select for those DEGs that are not responsive to OA-related changes with ongoing OA pathophysiology, we next prioritized DEGs that did not occur in the dataset of 2386 DEGs, as previously reported, to differ between preserved and lesioned OA cartilage [19]. This resulted in a total of N = 114 DEGs with same direction of effects in knee and hip but not differential in lesioned OA cartilage, which was considered for further analyses (genes are listed in Appendix A).

As shown in Figure 2A, proteins encoded by these genes have multiple functional interactions, which was significantly more than expected by chance (*P* = 2.0 × 10^−6^). Furthermore, significant enrichment for genes involved in biological processes regulating extracellular matrix remodeling (*P* = 2.6 × 10^−2^; nodes depicted in red) and ossification (*P* = 4.0 × 10^−2^; nodes depicted in blue) was found.

### 3.3. miRNA Expression Levels in Correlation with Expression of Genes Marking Early OA

To assess the correlation between the expression of miRNA and genes marking early OA in preserved OA cartilage, expression levels of *N* = 114 selected genes were reproduced from our previously established mRNAseq dataset [19] for which miRNA levels were here determined in plasma (*N* = 22). Upon determining within-subject correlations of selected 114 genes with 384 miRNAs robustly expressed plasma, 34 unique circulating miRNAs were identified that showed a strong correlation (ρ ≥ |0.7|) to 21 unique cartilage genes. Together, the proteins encoded by the 21 genes did not show significantly more protein–protein interactions than expected by chance (*P* = 8.0 × 10^−1^); however, we found significant enrichment for biological processes related to DNA-binding (Figure 2B, *P* = 2.7 × 10^−2^; nodes depicted in red) and Wnt-signaling (Figure 2B, *P* = 3.8 × 10^−2^; nodes depicted in blue).

As shown in Figure 3, the strongest correlation was observed between let-7d-5p and *EGFLAM* encoding EGF-like, Fibronectin type III and Laminin G domains (ρ = −0.75, *P* = 6.9 × 10^−5^). Additionally, *EGFLAM* strongly correlated with nine other miRNAs among which were several members of the let7-family (Table 2A and Appendix A). Notably, in addition to *EGFLAM*, three other genes significantly correlated with at least 10 miRNAs (Table 2A); *SMIM3*, encoding Small Integral Membrane Protein 3 (*N* = 17 miRNAs), *CTHRC1*, encoding Collagen Triple Helix Repeat Containing 1 (*N* = 14 miRNAs), and *HMGB2* encoding high-mobility group protein B2 (11 miRNAs). Additional exploration was performed to find whether any of the 21 unique cartilage genes were previously predicted and/or validated targets of identified unique and strongly correlating 34 plasma miRNAs. This showed that *HMGB2* is a validated target of miR-23b-3p, *SERBP1* a validated target of let-7e-5p, and *BTG2* a validated target of both miR-106b-5p and miR-132-5p; Appendix A).

### 3.4. Receiver Operator Curves with Selected Plasma miRNAs as Determinants of OA Progression

Next, the potential predictive value of miRNAs correlating with early OA genes was addressed. To that end, 34 most significant miRNAs were used, with strong correlation (ρ ≥ |0.7|; Appendix A) to 21 unique genes among which were the afore-mentioned 4 genes (*EGFLAM*, *SMIM3*, *CTHRC1*, *HMGB2*). Using Z scores, 7 miRNAs were determined to characterize progression over 2 years: miR-1307-5p, miR-140-3p, miR-181a-3p, miR-221-5p, miR-4326, miR-443, and miR-99a-5p (Appendix A). Discrimination between progressors and non-progressors was assessed by calculating the AUC with regression coefficients for selected plasma miRNAs while adjusting for sex, age, and BMI. In parallel AUC was calculated while only including sex, age, and BMI. Figure 4A shows strong increase toward clinically relevant AUC when including all 7 miRNAs as compared to covariates only (AUC = 0.86 versus 0.59, respectively; Appendix A shows boxplots for individual miRNAs with progression). Notably, when only the 4 most significant miRNAs were included (miR-1307-5p, miR-181a-3p, miR-4326, miR-4443), still a considerable predictive value was reached (AUC = 0.82), while progression over 5 years with these miRNAs could also still be distinguished with AUC = 0.75 (Figure 4B).

## 4. Discussion

By applying miRNA sequencing to plasma in parallel with RNA sequencing to articular cartilage of the same individuals from the RAAK study, we here showed that strong correlations exist between expression levels of circulating miRNAs and cartilage genes. In this way, we identified circulating miRNAs that strongly correlate with markers of early OA, thereby reflecting onset of OA pathophysiology. By applying identified miRNAs in an independent study cohort (GARP) including hip and knee OA, we showed clinically relevant predictive potential for 2-year and 5-year progression of hip or knee OA toward AUC ≥ 0.8, while additionally adding information about molecular pathways underlying the early OA pathology. We advocate that this signature of plasma miRNAs can contribute to distinguishing, at an early stage, individuals likely to develop OA, independent of the OA status.

Biological relevance of correlations between gene expression in tissues and miRNA levels in blood is still speculative. Certainly, our study does not determine the nature of the correlations, and it is not clear yet whether identified correlations are the cause or the consequence of the ongoing joint pathophysiology. Nevertheless, the fact that several of the genes marking early OA were identified in strong correlation with multiple plasma miRNAs may suggest that these genes, which are not strongly correlated among each other (Table 2B), mark important biological pathways (Figure 2) and require strict regulation and/or finetuning in addition to messaging in other tissues. Of importance, for that matter, is *HMGB2* (high-mobility group protein B2). Loss of *HMGB2* expression has been demonstrated to lead to senescence via induction of CTCF activity [39]. In OA cartilage, expression of *HMGB2* was downregulated as compared to non-OA cartilage [17,18], and it has been shown that HMGB2 regulates chondrocyte hypertrophy by mediating runt-related transcription factor 2 (*RUNX2*) expression and Wnt signaling [40]. Levels of *HMGB2* in cartilage were here identified in strong inverse correlation to levels of plasma miR-23b-5p, and we found that it was also previously validated as a target of miR-23b-5p. For that matter, levels of miR-23b were previously shown to be increased in equine synovial fluid early in OA [41]. Founded by the vision of OA as a whole joint disease, it is tempting to speculate that the initiating joint pathology involves signaling between the synovium and synovial fluid, the cartilage, and into the circulation. In addition to miR-23b, miR-99a was also shown to be increased in equine synovial fluid early in OA. Notably, miR-99a-5p, present in the miRNA signature, has been shown to affect macrophage polarization, a process shown to have critical impact on tissue repair and maintenance of tissue homeostasis [42], which are important factors in OA development.

Among the miRNA-signatures, we identified miR-140, which has been the subject of multiple studies in OA [11] and was identified previously as a potential OA biomarker [15]. Another miRNA with predictive value for OA development in our study was miR-1307-5p. It was recently shown that the suppression of miR-1307-5p results in increased TGFBI signaling, which promotes chondrocyte proliferation and inhibits apoptosis in OA mice [43]. Additionally, Nakamura and colleagues showed that miRNA-181a-5p antisense oligonucleotides have cartilage-protective effects, particularly for knee and facet joints [44]. Although in our study miR-181a-5p was not amongst the 34 miRNAs with ρ ≥ |0.7| and thus not in the miRNA signature for OA progression (Appendix A), we did find miR-181a-5p levels with moderate correlation to *GLI3* expression levels in cartilage (ρ = −0.6, *P* = 2.1 × 10^−3^; Appendix A). The described upregulation of *GLI3* in OA cartilage with the here-observed negative correlation would indeed be in line with a potential beneficial role for miRNA-181a-5p in cartilage homeostasis. Finally, miR-221-5p, previously found to be upregulated in lesioned as compared to preserved OA cartilage [19], and now shown to have predictive potential for OA progression, was shown to be induced in response to mechanical loading of bovine and mouse cartilage [45].

The strength of our study is the combined approach in the RAAK and in the GARP study, thereby allowing analyses of the same circulating miRNAs in two independent study cohorts while including data from non-endstage OA patients. Previous studies have explored the potential of circulating miRNAs as biomarkers in OA [11,46,47]. However, many of such studies have taken a targeted approach. More in line with our approach was the approach from Beyer and colleagues demonstrating, for the first time, the potential of miRNAs in the circulation as biomarkers in OA. Using microarrays, they showed that let-7e level was a negative predictor for total joint arthroplasty within 2 years [14]. Although members of the let-7 family were not among the 7 miRNA signature determined in the current study, we did observe a strong correlation for levels of multiple family members (ρ > |0.6| for let-7a-5p, let-7d-5p, let-7e-5p, let-7f-5p, and let-7g-5p) to the expression of early markers, specifically to *EGFLAM*. More recently, Ali et al. performed miRNA sequencing to identify a 7-miRNA-signature of early knee OA patients (miR-191-3p, miR-199a-5p, miR-335-3p, miR-335-5p, miR-543, miR-671-3p, and miR-1260b) [16]. Possibly, as a result of the different approach we took, aiming to identify a panel that could serve for both hip and knee OA, there was no overlap with the miRNAs identified in the current study. Since miR-191-3p moderately correlated to age (ρ = 0.51, *P* = 1.6 × 10^−2^; Appendix A), this miRNA was disregarded in our analysis. Of the other miRNAs of the signature, interestingly, the strongest correlation in our datasets was observed for miR-335-5p and *HMGB2* (ρ = −0.61, *P* = 2.8 × 10^−3^). None of the other miRNAs were correlated with at least ρ = |0.6| to the early markers of OA.

The drawback of our study is the fact that analyses of cartilage gene expression and OA progression cannot be addressed in a longitudinal study cohort and requires different types of studies. Furthermore, our sample size may have precluded the identification of lowly expressed miRNAs with predictive value, although clinical applicability of such lowly expressed miRNAs could be questioned. Finally, our study cohorts showed heterogeneity with majority of participants being female. Although we did not find a significant association of the plasma miRNAs with sex ( Appendix A) nor clustering for sex in the plots (Figure 3), we cannot exclude the fact that this may have introduced some bias in our results and that the identified signature is more effective for females.

## 5. Conclusions

Taken together, we show that plasma miRNA expression levels correlate to gene expression levels in cartilage and suggest that this can be exploited to represent ongoing pathophysiological processes in articular cartilage. As such, in the expression levels of a panel of 7 circulating microRNAs, each one individually correlates with gene expression levels of early OA markers in cartilage, together with the potential for clinically relevant predictive value for progression of osteoarthritis over 2 and over 5 years. Following the current strategy, demonstrating correlations between molecular changes in joint tissue and plasma miRNA levels, it is tempting to speculate that circulating miRNAs may also serve to identify subtypes of OA [48]. However, this remains to be established. Currently, replication of identified miRNAs in additional follow-up population-based cohorts is needed to confirm whether application of the plasma miRNA panel could provide a window of opportunity to identify patients prone to develop OA.

## Figures and Tables

**Figure 1 biomolecules-11-01356-f001:**
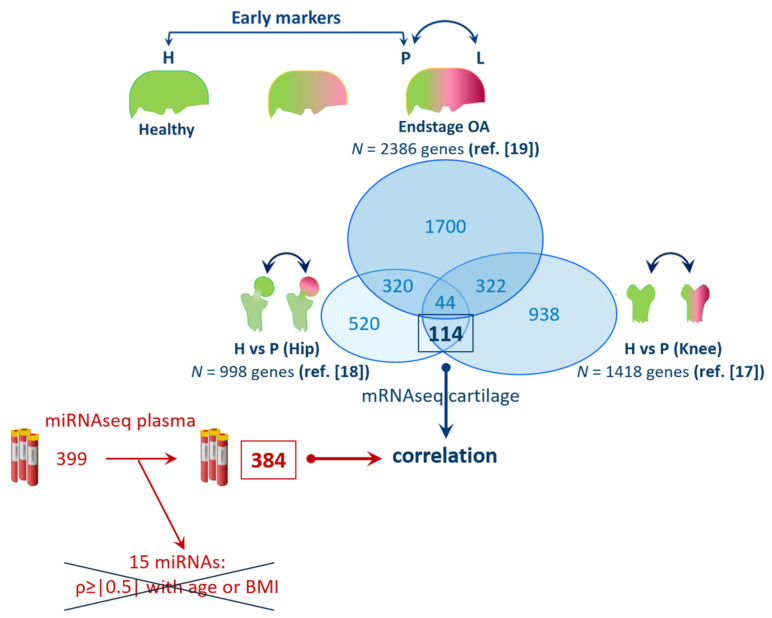
Selection of 114 genes marking early changes in articular cartilage and correlating to levels of 384 plasma miRNAs (Legend—H: healthy cartilage; P: preserved; and L: lesioned OA cartilage; ref.: references used in this paper for the selection of genes; 15 miRNAs were disregarded due to their potential correlation ρ ≥ |0.5| with age and BMI).

**Figure 2 biomolecules-11-01356-f002:**
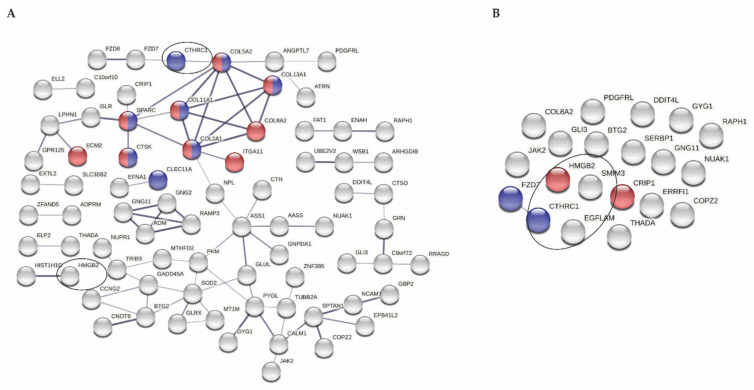
Protein–protein network. (**A**) STRING analysis for 114 genes included in whole correlation analyses; indicated in red and blue are significantly enriched pathways (extracellular matrix and ossification, respectively; disconnected nodes are hidden). (**B**) STRING analysis for 21 unique genes strongly correlating to 34 miRNAs; ρ ≥ |0.7|; indicated in red and blue are significantly enriched pathways (DNA binding and Wnt-protein binding, respectively). Circles point at the 4 genes strongly correlating to levels of at least 10 different plasma miRNAs (*EGFLAM* and *SMIM3* are disconnected in (**A**)). Note: LINC00115 is missing because this is a non-coding RNA.

**Figure 3 biomolecules-11-01356-f003:**
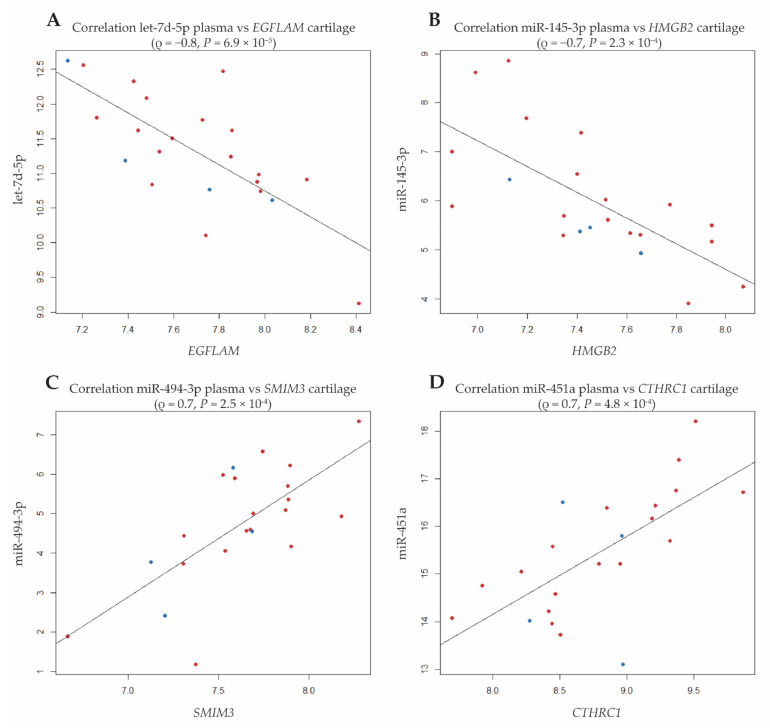
Dot plots of strongest correlations between the 4 cartilage genes strongly correlating to levels of at least 10 different plasma miRNAs (pointed out in Figure 2: (**A**): *EGFLAM*; (**B**): *HGMB2*; (**C**): *SMIM3*; (**D**): *CTHRC1*; red dots refer to female and blue dots to male samples).

**Figure 4 biomolecules-11-01356-f004:**
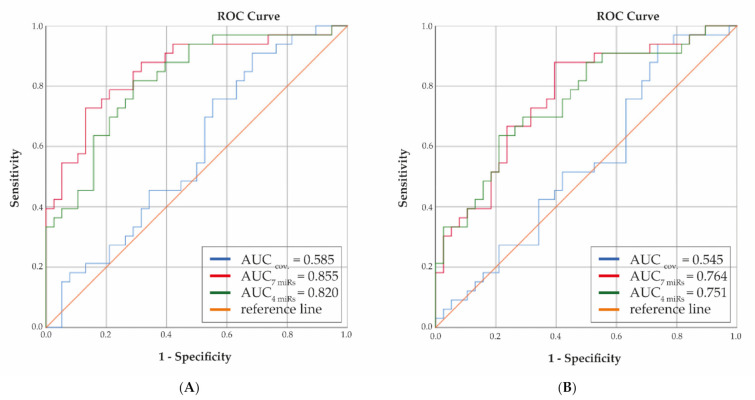
Receiver Operator Curve predictive value of selected plasma miRNAs at 2 (**A**) and 5 years (**B**) follow-up (sex, age, and BMI only: blue line; including 7 miRNAs (miR-1307-5p, miR-140-3p, miR-181a-3p, miR-221-5p, miR-4326, miR-4443, miR-99a-5p): red line; including 4 miRNAs (miR-1307-5p, miR-181a-3p, miR-4326, miR-4443): green line).

**Table 1 biomolecules-11-01356-t001:** Sample characteristics RAAK (**A**) and GARP (**B**); (non-prog: non-progressors; prog: progressors).

**A**		
Sex	18/22 Female	
Age	55–81 (avg. 71.1)	
BMI	21–33 (avg. 27.9)	
Joint	16/22 Knee	
**B**		
	Non-Prog.	Prog.
Sex	33/38 Female	29/33 Female
Age	47–75 (mean: 61.4)	50–69 (mean: 59.4)
BMI	20–34 (mean: 26.1)	20–40 (mean: 26.2)
Joint	18/38 Knee	17/33 Knee

**Table 2 biomolecules-11-01356-t002:** (**A**) Unique genes showing strong correlation with plasma miRNAs (ρ ≥ |0.7|, *P* < 6.9 × 10^−5^). Additional miRNAs with ρ ≥ |0.6| and direction of differentially expressed genes (DEG; preserved OA cartilage versus healthy cartilage) are shown. (**B**) Correlations (left) and matrix (right) of genes strongly correlating to at least 10 plasma miRNAs (indicated in bold in (**A**).

**A**						
**miRNA**	**Gene**	**Corr.**	** *P* **	**Additional miRNAs (ρ ≥ |0.6|)**		**Cartilage** **DEG** **(P vs. H)**
let-7d-5p	** *EGFLAM* **	−0.75	6.8 × 10^−5^	let-7f-5p; let-7a-5p; miR-4443; miR-221-5p; miR-3615; miR-200b-3p; let-7e-5p; miR-1180-3p; let-7g-5p		dn
miR-3928-3p	*PDGFRL*	−0.74	9.0 × 10^−5^	miR-1260a; miR-106b-5p; 6852-5p; miR-23b-5p		up
let-7a-5p	*THADA*	−0.73	1.0 × 10^−4^	miR-339-3p; miR-22-3p		dn
miR-145-3p; miR-23b-3p	** *HMGB2* **	−0.71	2.3 × 10^−4^	miR-181a-3p; miR-425-3p; miR-7849-3p; miR-326; miR-339-5p; miR-133a-3p; miR-3613-5p; miR-335-5p; miR-421		dn
miR-19b-3p	*GLI3*	−0.71	2.4 × 10^−4^	miR-3909; miR-23b-5p; miR-181a-5p; miR-4755-5p		up
miR-494-3p	** *SMIM3* **	0.71	2.5 × 10^−4^	miR-889-3p; miR-411-3p; miR-224-5p; miR-379-5p; miR-4326; miR-222-3p; miR-431-5p; miR-12136; miR-382-5p; miR-329-3p; miR-495-3p; miR-505-3p; miR-221-5p; miR-7849-3p; miR-30b-3p; miR-6772-3p; miR-493-5p; miR-381-3p		dn
miR-106b-5p	*COPZ2*	0.70	2.7 × 10^−4^	miR-939-3p; miR-210-3p		up
**B**						
**Gene**		**Gene**		**Corr.**		** *P* **
*HMGB2*		*SMIM3*		−0.39		7.0 × 10^−^^2^
*CTHRC1*		*SMIM3*		−0.31		1.7 × 10^−^^1^
*CTHRC1*		*EGFLAM*		0.22		3.2 × 10^−^^1^
*CTHRC1*		*HMGB2*		0.22		3.3 × 10^−^^1^
*EGFLAM*		*HMGB2*		0.19		4.0 × 10^−^^1^
*EGFLAM*		*SMIM3*		−0.10		6.5 × 10^−^^1^
		** *CTHRC1* **		** *EGFLAM* **	** *HMGB2* **	** *SMIM3* **
*CTHRC1*		1.00				
*EGFLAM*		0.22		1.00		
*HMGB2*		0.22		0.19	1.00	
*SMIM3*		−0.31		−0.10	−0.39	1.00

## Data Availability

All data from the study are available by Public Access within the text or online.

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
