# Peer review of "Circulating MicroRNAs Highly Correlate to Expression of Cartilage Genes Potentially Reflecting OA Susceptibility—Towards Identification of Applicable Early OA Biomarkers"

_biomolecules, 2021, doi:10.3390/biom11091356_

Round 1

Reviewer 1 Report

The crucial findings the manuscript, that is that the levels of seven plasma circulating miRNAs identified in OA patients could be correlated with the levels of early-OA transcripts is potentially interesting. I find this story very interesting and the attempts to identify molecular biomarkers of early OA is definitely needed, especially using high-throughput technologies, like presented here RNAseq. I also strongly believe that “(…) application of the plasma miRNA panel could provide a window of opportunity to identify patients prone to develop OA.” However, I cannot find the results presented here as convincing and strong statements like in the title should be definitely restrained.

My main criticism concerns the selection of the study group. I know this group was already previously subjected to RNAseq and this is cool to have mRNA data, but I am not convinced that including patients at really old age (av. 71!) and therefore possibly already advanced OA, would reflect early OA miRNAs. You did excluded miRNAs correlated with age but still don’t you think that you would rather identify miRNAs expressed in late OA? In the best case, a portion of them would be late-OA miRNAs. (In that respect, selection of mRNAs marking early OA was done properly.) Moreover, the patients characteristics presented here is very limited. Did all 22 patients suffer from early OA? What about KL scale?

Using RNAseq you would definitely identify a plethora of miRNAs. But not having a control (healthy) group of patients means that you identify not miRNAs differentially expressed and connected with OA. They could be expressed also in healthy individuals, right? And this is what you definitely don’t want as a potential biomarker. Healthy miRNAs could also correlate with early OA mRNAs and you omitted that in your analysis.

Heterogenicity of the patients: 18 female patients and 4 male make the group unequal and could also provide some bias.

34 miRNAs correlated with 21 mRNAs – are there the possible targets of identified miRNAs among them?

ROC curves are impressing, but I’m still afraid that healthy miRNAs are among these 7 the coolest predictive ones.

To date, it has been reported that about 80 miRNAs – please discuss the previous reports on these 7 miRNAs you found to be potentially predictive.

Due to not having a control group, some of the sentences have to be smoothened or deleted – their message is to strong to the evidence, e.g:

“We advocate that this signature of plasma miRNAs can contribute to distinguish at an early stage individuals likely to develop OA, independent of the OA status”

“Taken together, we show that plasma miRNAs reflect gene expression levels in cartilage and can be exploited to represent ongoing pathophysiological processes in articular cartilage.”

Some other points:

  • miRNAs do not correlate to genes, rather their expression levels or the appearance
  • “Previous studies have explored the potential of circulating miRNAs as biomarkers in OA” – please cite here some relevant references, at least the newest reviews
  • The Material and methods section is too concise. Therefore I am not sure if you took into consideration up- or down-regulated miRNAs. What was their abundance? You coul present a table in Supplementars
  • If would be of crucial relevance if you would validate your RNAseq results with e.g. real-time PCR or so on

Author Response

Reply to the reviewers:

We sincerely thank the reviewer for the time, effort, and insightful comments identifying the weaknesses in our manuscript, and providing us with the opportunity to revise and strengthen our study. The points raised from the original manuscript have been addressed to the best of our abilities and we think it has substantially improved it (for the reviewers convenience, our response is also uploaded as PDF-document: Please see the attachment).

General comments:

  1. The reviewers coincide in the suggestion that ‘Several statements in the discussion went beyond the scope of this study’ and ‘strong statements like in the title should be definitely restrained’.

Answer: We agree to the reviewer that the statement put forward in the title is premature for the data we show.

Author Action: We have now changed our title to ‘Circulating microRNAs highly correlate to expression levels of cartilage genes potentially reflecting OA susceptibility - towards identification of applicable early OA biomarkers’. This, to highlight the correlation between expression levels of plasma miRNAs and cartilage genes. Furthermore, we have edited the abstract in line with the first paragraph of the manuscript by concluding that ‘We advocate that identified signature of 7 plasma miRNAs can contribute to direct further studies towards early biomarkers predictive for progression of osteoarthritis over 2 and 5 years’. Also, we have edited the last section to more carefully phrase our conclusions (lines 367-370: ‘Currently, replication of the miRNAs in additional follow-up population based cohorts is needed to further affirm our conclusions that application of the plasma miRNA panel could provide a window of opportunity to identify patients prone to develop OA’).

  1. Please note that, when updating Figure 3 to the new format indicating male and female samples as mentioned in our response to comment 4 of Reviewer 1, we observed that axes in the original figure were inversed. We apologize for this oversight. It does not affect the results nor our conclusions and has been corrected in the new Figure 3.

Further points raised by the reviewers (in italics) and our answers here below, with any page and line numbers referring to the revised manuscript without tracked changes:

REVIEWER 1:

  1. The reviewer states that ‘…cannot find the results presented here as convincing and strong statements like in the title should be definitely restrained.

Answer: We agree to the reviewer as mentioned and responded above in ‘General comments’ (nr. 1). However, we do not agree with the observation that our results are little convincing. In contrast, we ourselves are confident that the highly significant and strong correlations of identified expression levels for plasma miRNA and cartilage mRNA (e.g. let-7d-5p and EGFLAM with ρ=-0.75, P=6.9x10-5) are robust which is further confirmed and reflected in the correlation plots presented in the manuscript (Figure 1). Moreover, we consider the fact that the expression levels of identified plasma miRNAs in the RAAK study also showed high predictive value for progression of OA as determined with ROC in another (independent!) study (GARP) is actually supporting and adding value to our findings.

  1. My main criticism concerns the selection of the study group [...] I am not convinced that including patients at really old age (av. 71!) and therefore possibly already advanced OA, would reflect early OA miRNAs’.

Answer: The reviewer may have misunderstood the very strict selection criteria that we took for the identification of early markers (i.e. gene expression levels in preserved cartilage). This approach was highlighted in the second paragraph of the result section (lines 187-197) and Figure 1. To reiterate, we have taken an in silico approach to pre-select genes from previously published datasets. In step 1 we selected differentially expressed genes reported in studies from Karlsson et al (knee OA) and from Xu et al (hip OA; respectively references 17 and 18 of our manuscript) that compared non-OA cartilage to preserved OA cartilage (i.e. cartilage from the macroscopically intact area of OA joints). As such, selection represents genes that mark or predispose to OA onset. In step 2 we further selected for genes that were not differentially expressed between preserved and lesioned OA cartilage of the same joint (reference 19) thus: genes not responsive to OA status of the cartilage tissue. For that matter, although preserved cartilage of older OA patients were included, levels of selected genes and associated miRNAs are independent of changes in gene expression due to age and/or OA status of the tissue. Additionally, we consider, as mentioned in the discussion, ‘Strength of our study is the combined approach in the RAAK and in the GARP study, thereby allowing analyses of the same circulating miRNAs in two independent study cohorts while including data from non-end-stage OA patients’ (lines 327-329 of the discussion).

Having said all this, we acknowledge in the discussion that, to avoid false positive results, our identified plasma miRNAs need to be confirmed in additional follow-up population based cohorts to further affirm our conclusions.

Author action: Since it is very important for the reader to understand characteristics of genes that we have selected as early markers, we have now included fold difference and Pvalue for gene expression in preserved and lesioned OA cartilage. For genes included in correlation analyses (Supplementary Table S2A, N=114) as well as for the genes that were disregarded due to their significant differential expression within the preserved and lesioned areas of OA cartilage (Supplementary Table S2B, N=44).

  1. 3.Using RNAseq you would definitely identify a plethora of miRNAs. But not having a control (healthy) group of patients means that you identify not miRNAs differentially expressed and connected with OA. They could be expressed also in healthy individuals, right?

Answer: Given the unique feature of circulating miRNAs to be able to sensitively reflect (patho)physiological processes of tissues, they have been recognized as a powerful new biomarker source. In response to the suggestion from this reviewer (‘Some other points’ B) we now refer to three recent reviews where this has been discussed (references 41-43 of the revised manuscript).

Having established within person RNA sequencing datasets of gene expression in (OA) joint tissues, as well as of miRNA in the circulation, we are in an ideal situation to exploit this novel biomarker source for possible application in the OA field. Henceforth, the major aim of our study was to identify miRNAs in the circulation that correlated to preselected genes expressed in preserved cartilage. So indeed, the plasma miRNAs identified here are likely also expressed in healthy individuals. Even more, healthy individuals with the indicated specific signature of miRNA expression (i.e. those with early OA marker gene expression levels in cartilage) are susceptible to OA onset. As such, we aimed in our manuscript at detecting a specific pattern in concurrent expression of multiple miRNAs (the 7-miRNA panel) that together are indicative of having a propensity to develop OA (Figure 4).

  1. Heterogenicity of the patients: 18 female patients and 4 male make the group unequal and could also provide some bias’.

Answer: Given the fact we have assessed within-patient correlation the inclusion of male participants should in principle not have influenced our result. Moreover, despite potential heterogeneity we aim to identify general applicable markers. Having said this, we acknowledge that age, sex, and BMI could be important predictors of OA onset, hence in our ROC analyses we have adjusted for these covariates (see Figure 4).

Author action: We have now commented on this in the discussion (lines 352-357), distinguish miRNAs expressed in plasma of females (red dots) and of males (blue dots; Figure 3), and have added Supplementary Table S1B with Pvalues for association between miRNA levels and sex in the RAAK study (lowest Pvalue P=8.5x10-2 for miR-4443 being none significant).

  1. 34 miRNAs correlated with 21 mRNAs – are there the possible targets of identified miRNAs among them?

Answer: A very interesting question: yes, there were three potential targets. However, we preferred not to extensively address this in the current study since we focused on the correlation towards early markers for OA progression. Especially, it is not clear whether identified correlations are causal or consequential, and for the correlation in tissues and blood this is still speculative.

Author action: We have now shortly commented on this in the manuscript (lines 234-238) and present an additional Supplementary Table (S4).

  1. ROC curves are impressing, but I’m still afraid that healthy miRNAs are among these 7 the coolest predictive ones.

Answer: We thank the reviewer for this compliment, and think indeed our study will contribute to further direct identification of early markers. However, we are not entirely sure what the reviewer suggests other than that commented in nr. 3? Indeed we need to establish the predictive value of the identified miRNAs in a prospective population based study. An important step to further assess specificity and sensitivity of our miRNAs. This will also show which miRNAs are protective (i.e. expressed in healthy participants) and which may be detrimental (higher expressed in patients). However, we consider this outside the scope of the current study.

  1. To date, it has been reported that about 80 miRNAs – please discuss the previous reports on these 7 miRNAs you found to be potentially predictive.

Answer: We have now added a section to the discussion where we discuss previous publications regarding the identified miRNAs.

  1. Due to not having a control group, some of the sentences have to be smoothened or deleted – their message is too strong to the evidence, e.g: “We advocate that this signature of plasma miRNAs can contribute to distinguish at an early stage individuals likely to develop OA, independent of the OA status” and “Taken together, we show that plasma miRNAs reflect gene expression levels in cartilage and can be exploited to represent ongoing pathophysiological processes in articular cartilage.

Answer: We have now adapted the sentences as indicated in the general part of the response here above.

Some other points:

A          ‘miRNAs do not correlate to genes, rather their expression levels or the appearance’:

Author action: The specification has now been introduced in the revised manuscript (e.g. abstract line 24; results section 3.3. lines 216-217; discussion line 279).

B          “Previous studies have explored the potential of circulating miRNAs as biomarkers in OA” – please cite here some relevant references, at least the newest reviews: Answer: This was indeed an omission in the original manuscript and references have now been added.

C          ‘The Material and methods section is too concise. Therefore I am not sure if you took into consideration up- or down-regulated miRNAs. What was their abundance? You could present a table in Supplementars’: Author action : We have now provided more detailed information in Supplementary Figure S1 to show difference in miRNA levels between patients with minimal progression and OA progressors. This is mentioned in line 264. For the reviewers information we here below show a table with the BaseMean and VST-values of the miRNAs in our signature.

miRNA

BaseMean

VST-value

miR-1307-5p

31.67

4.71

miR-140-3p

5222.77

11.72

miR-181a-3p

21.92

4.27

miR-221-5p

62.2

5.45

miR-4326

37.62

4.51

miR-4443

54.3

5.36

miR-99a-5p

2772.9

11.49

Table 1. BaseMean and VST-values of miRNAs for identified miRNAs in our study.

D          ‘If would be of crucial relevance if you would validate your RNAseq results with e.g. real-time PCR or so on’:

Answer: RNA sequencing is inherently more sensitive than PCR, and there is some discussion in the field as to whether RNA sequencing needs validation by PCR at all. This is demonstrated for example in a paper by Everaert et al (2017, doi:10.1038/s41598-017-01617-3) and exemplified in a recent paper by Coenye (2021, doi:10.1016/j.bioflm.2021.100043) stating ‘While microarrays allowed to carry out gene expression studies on a scale not seen before, and despite their overall high level of performance, some concerns were raised about reproducibility and bias. […] However, RNA-seq does not suffer from the same issues as (some) microarrays did and there are a number of studies that have specifically addressed the correlation between results obtained with RNA-seq and qPCR’. Furthermore, we considered it may be more informative to replicate our findings now in another cohort we have not validated results here.

Author action We have now addressed the latter in our discussion (line 367-370). However, we will be happy to perform qPCR in case the reviewer insists.

Reviewer 2 Report

Ramos and colleagues investigated the expression of circulating miRNAs in OA patients. They identified several interesting miRNAs that theoretically affect important genes in OA pathology. In addition, they found that candidate miRNAs linked to OA progression. They suggested that miRNA panels may be useful as a predictive tool for OA progression.

Their results emphasize the importance of analyzing miRNAs in plasma in OA patients and this study is important if the future studies address the crucial role of such miRNAs for OA pathology. I would raise following criticisms.

  1. The number of RAAK study is too small and thus it does not give enough statistical power.
  2. They applied selected miRNA data obtained from initial study to another GARP study. However, they did not confirm that these selected miRNAs were in fact significantly changed in plasma in GARP study.
  3. Line 176-180: what do these sentences mean?     
  4. Figure 2 is difficult to understand because of poor quality.
  5. Several statements in the discussion went beyond the scope of this study. The authors may need to reconsider in order to match what the data actually shows.

Author Response

Reply to the reviewer:

We sincerely thank the reviewer for the time, effort, and insightful comments identifying the weaknesses in our manuscript, and providing us with the opportunity to revise and strengthen our study. The points raised from the original manuscript have been addressed to the best of our abilities and we think it has substantially improved it (for the reviewers convenience, our response is also uploaded as PDF-document: Please see the attachment).

General comments:

  1. The reviewers coincide in the suggestion that ‘Several statements in the discussion went beyond the scope of this study’ and ‘strong statements like in the title should be definitely restrained’.

Answer: We agree to the reviewer that the statement put forward in the title is premature for the data we show.

Author Action: We have now changed our title to ‘Circulating microRNAs highly correlate to expression levels of cartilage genes potentially reflecting OA susceptibility - towards identification of applicable early OA biomarkers’. This, to highlight the correlation between expression levels of plasma miRNAs and cartilage genes. Furthermore, we have edited the abstract in line with the first paragraph of the manuscript by concluding that ‘We advocate that identified signature of 7 plasma miRNAs can contribute to direct further studies towards early biomarkers predictive for progression of osteoarthritis over 2 and 5 years’. Also, we have edited the last section to more carefully phrase our conclusions (lines 367-370: ‘Currently, replication of the miRNAs in additional follow-up population based cohorts is needed to further affirm our conclusions that application of the plasma miRNA panel could provide a window of opportunity to identify patients prone to develop OA’).

  1. Please note that, when updating Figure 3 to the new format indicating male and female samples as mentioned in our response to comment 4 of Reviewer 1, we observed that axes in the original figure were inversed. We apologize for this oversight. It does not affect the results nor our conclusions and has been corrected in the new Figure 3.

Further points raised by the reviewers (in italics) and our answers here below, with any page and line numbers referring to the revised manuscript without tracked changes:

REVIEWER 2:

  1. The number of RAAK study is too small and thus it does not give enough statistical power.’

Answer: We regret that the reviewer thinks the inclusion of 22 RAAK patients is too small. However, the reviewer may not have understood that correlations between cartilage gene expression levels and circulating miRNA levels were performed within individuals which has high power and is independent of other baseline characteristics such as age, sex, OA status, and BMI. For that matter, our results (Table .2A and Supplementary Table S3) show highly significant and strong correlations (ρ≥|0.7|). As such we are confident to have identified robust results. Nevertheless, increasing sample sizes could have resulted in the identification of additional (e.g. low expressed) miRNAs. Furthermore, we want to stress that apart from the 22 RAAK participants, the analysis was further continued in 71 participants of the GARP study, with strong predictive value. This supports the value of our analysis.

Author action: We have now added in the discussion that our sample size may have precluded the identification of lowly expressed miRNAs with good predictive value (lines 350-352), although we also question whether these are the most clinically relevant miRNAs to serve as biomarkers.

  1. They applied selected miRNA data obtained from initial study to another GARP study. However, they did not confirm that these selected miRNAs were in fact significantly changed in plasma in GARP study.

Author action: We have now added Supplementary Figure S1 in which we show boxplots for the expression of miRNAs in both progressors and non-progressors.

  1. Line 176-180: what do these sentences mean?

Answer: We are not completely sure but we think the reviewer is referring to the figure legend.

Author action: We have now adapted, both Figure 1 and the legend, which will hopefully indicate more clearly the step-wise procedure we took for the selection of cartilage genes changing early in OA.

  1. Figure 2 is difficult to understand because of poor quality.

Author action : We have increased the quality and hided disconnected proteins to simplify the image.

  1. Several statements in the discussion went beyond the scope of this study. The authors may need to reconsider in order to match what the data actually shows.

Answer: We agree to the reviewer as mentioned and responded in the ‘General comments’ (nr. 1).

Round 2

Reviewer 1 Report

The Authors presented very nice work. All of my comments are addressed 9changes in the manuscript or at least discussed). I'm still not convinced abut this no-healthy-control group, but you did a goos job when revising the paper. I'd reccomend it for accepting.

Reviewer 2 Report

no comments